# Effect of forest cover change on the prevalence of acute respiratory tract infections, diarrhoea, and fever among children under five: Using an ecosystem approach to child health

Amit Timilsina[1]*, Binaya Chalise[2], Kedir Y. Ahmed[3,4], Subash Thapa[3,5]

1 Research and Community Development Center, Kathmandu, Nepal, 2 Network for Education and Research on Peace and Sustainability, Hiroshima University, Hiroshima, Japan, 3 Rural Health Research Institute, Charles Sturt University, Orange, New South Wales, Australia, 4 Translational Health Research Institute, Western Sydney University, Campbelltown Campus, New South Wales, Australia, 5 Health Inequality and Resilience Research Institute, Kathmandu, Nepal

* timilsinaamit@gmail.com

## Abstract

### Introduction

An ecosystem approach to child health emphasizes the interconnectedness of environmental, social, and biological factors in shaping children's health and well-being. However, it is not known whether changes in forest cover have an effect on common childhood illnesses. This study investigated the associations between forest cover and acute respiratory infections (ARI), diarrhoea, and fever among children under five in Nepal.

### Method

This was a cross-sectional study based on the analysis of the Nepal Demographic and Health Surveys (NDHS) datasets of 2011 (N = 5054) and 2016 (N = 4861). Forest cover data for the years 2011 and 2016 were extracted from high-resolution raster images from NASA Earthdata (30m resolution). We employed a logit model on the geo-linked NDHS datasets to compute the marginal effect of forest cover on ARI, diarrhoea, and fever.

### Results

From 2011 to 2016, the prevalence of fever increased from 18% to 20%, while the prevalence of diarrhoea decreased from 14% to 7%, and ARI prevalence decreased from 5% to 3%. The mean tree cover percentage also decreased from 21% in 2011 to 19% in 2016. Forest cover was significantly associated with reduced likelihood of diarrhoea symptoms among children in both 2011 and 2016. Change in forest cover between 2011 and 2016 was significantly associated with a reduced probability of diarrhoea by 3.39% (Δy/Δx: −0.0339, 95% CI: −0.0141, −0.0535; p-value: 0.001), after

**Data availability statement:** All relevant data are within the manuscript.

**Funding:** The author(s) received no specific funding for this work.

**Competing interests:** The authors have declared that no competing interests exist.

adjusting for all other variables. No significant associations were found between forest cover change and changes in ARI and fever prevalence among children under five.

## Conclusion

The present results should be interpreted cautiously, as they may not accurately reflect individual-level dynamics regarding the effect of forest cover on child health outcomes. The effect of forest cover in reducing childhood diarrhoea underscores the need for comprehensive child health programs that also incorporate environmental components.

## Introduction

Sustainable Development Goals (SDG) 3.2 aims to reduce under-5 mortality to fewer than 25 per 1,000 live births by 2030 [1]. There has been some progress, with premature death among children declining by 19% for diarrhoea and 64% for respiratory infections between 2000–2019 globally [2]. However, diarrhoeal and respiratory diseases remain among the top five leading causes of premature death among children globally.

An ecosystem approach to child health emphasizes the interconnectedness of environmental, social, and biological factors in shaping children's health and well-being [3]. Ecological factors significantly contribute to childhood illnesses and mortality, with a 15% of these factors attributed to water and sanitation issues, and an additional 8% to air pollution [4]. Global forest cover, which spans 4 billion hectares accounting for 30% of total land area, has declined by nearly nearly 10% (386 million hectares) over the past two decades [5]. Protecting and restoring forests is essential to combat climate change and its impacts on human health, including the prevention of major infectious diseases [6–8] and improvements in child health and nutrition [9].

Forests play a vital role in maintaining ecological balance by improving air quality, regulating microclimates, and reducing waterborne disease risks through their influence on water filtration and availability [10,11]. For instance, forest cover can mitigate air pollution, which is a key contributor to respiratory infections, and help regulate local climates, reducing heat-related illnesses that may exacerbate diarrhoea and fever [12]. Although some studies have found positive results regarding the health benefits of afforestation, including improved mental health, stress reduction, and clinical outcomes, other studies have noted mixed evidence [13,14].

Nepal has effectively managed its forest cover over the past two decades through successful community-based forest management systems [15]. This model has gained international recognition for its positive impact on environmental conservation and local development. Nepal's forest cover declined at a rate of 0.96% annually from 1976 to 1991 but then increased at a rate of 0.63%, reaching approximately 44.7%of the total land area, including both forests and other wooded lands, in 2015 [15]. The current forest cover stands at 4.8 million hectares, or 33% of the total land area, based on more recent national estimates, though the country has experienced net forest loss and fragmentation in specific geographic areas due to land-use change and infrastructure development [16].

Understanding the physiological and ecological mechanisms by which forest cover influences common childhood illnesses provides an evidence-based rationale for integrating forest conservation into public health strategies. This study, therefore, aimed to explore the impact of forest cover changes on the likelihood of childhood infections including acute respiratory infections (ARI), diarrhoea, and fever, among children under five.

## Methods

### Study design and data sources

This cross-sectional study was based on the analysis of the Nepal Demographic and Health Survey (NDHS) datasets for the years 2011 (N = 5,054) and 2016 (N = 4,861). The NDHS used a two-stage stratified cluster sampling design to select the study participants, that included first administrative units (e.g., regions) as urban and rural strata. In the first stage, Enumeration Areas (EAs) were randomly selected based on their size and a complete census of households was conducted in each selected EA. In the second stage, a systematic random sampling approach was employed to select a fixed number of households from each EA with equal probability.

The NDHS 2011 had 289 clusters from 13 sub-regions with 10,826 household interviewed, 12,674 eligible women and 5,054 under-5 children included. The NDHS 2016 had 383 clusters from 7 provinces with 11,040 households interviewed, 12,862 eligible women and 4,861 under 5 children included. Data were obtained from eligible women, who were defined as all females aged between 15 and 49 years who either lived permanently or were visiting the households on the night before the survey. The present study was restricted to the youngest children aged 0–59 months who resided with the respondent, yielding a weighted total of 5054 children in 2011 and 4861 children in 2016.

For each DHS cluster, a forest cover information was extracted from geospatial analysis of high-resolution raster images of global forest cover [17–20]. Due to displacement (geo-masking) of GPS coordinates for DHS clusters by 10 km for rural clusters and 2 km for urban clusters, circular buffers of 2 km for urban points and 10 km for rural points were used during data extraction of forest cover data [21]. The NDHS data was merged with Geographic Information System (GIS) tools that gathered data on tree cover raster available from NASA Earthdata Search (30m resolution) using satellite images. The raster was available for 2011 and 2016, and was based on Global Land Survey Data, Digital Elevation Model and MODIS VCF tree cover layer [19,20]. Other GIS covariates were retrieved from the DHS GIS data. All GIS operations were carried out in ArcGIS Pro 2.8 (*Esri, Redlands, CA, USA*) [22].

### Variables

The outcome variables included ARI, diarrhoea, and fever, which were measured by asking the mothers to recall if their child had fever, respiratory symptoms and diarrhoea within past two weeks preceding the survey (see Table 1). ARI symptoms included having had either cough, difficulty in breathing, or shortness of breath in the past 2 weeks preceding the survey. ARI was grouped as '1' = 'experienced ARI' or '2' = 'did not experience ARI'. Diarrhoea was defined as a passing of abnormal stools during the two weeks preceding the survey. Diarrhoea was grouped as '1' = 'experienced diarrhoea' or '2' = 'did not experience diarrhoea'.

The explanatory variable was forest cover which was defined as tree cover percentage within DHS buffer of 2 km for urban and 10 km for rural buffer. The covariates included in the analysis were selected based on their theoretical relevance to the ecosystem approach, evidence from prior literature, and data availability [3,12,14,23,24]. These included: mother-child characteristics, household general characteristics, and household environmental characteristics. The mother-child characteristics included child's age and sex, mother's education, access to media, birth order and birth size. The household general characteristics included the following: place of residence (urban/ rural), ethnicity (advantaged/disadvantaged), religion (Hindus/non-Hindus), ecological region (Mountain, Hill, and the Terai), family size, and wealth index. Due to the socioeconomic status and cultural hierarchy of these castes and ethnicities in Nepalese society, the DHS

**Table 1. Study variables.**

| Outcome variables | |
|---|---|
| Fever | Had fever in the past 2 weeks preceding the survey |
| Respiratory symptoms | Had cough difficulty in breathing and shortness of breath in the past 2 weeks preceding the survey |
| diarrhoea | Had diarrhoea in the past 2 weeks preceding the survey |
| **Explanatory Variables** | |
| Forest Cover | Tree cover percentage within Demographic health survey (2011,2016) buffer of 2 km for urban and 10 km for rural buffer |
| **Control Variables** | |
| Household (HH) Characteristics | Place of residence, ethnicity, religion, family size, own animal, wealth index and ecological region |
| Mother & child (MCH) characteristics | Child age, child sex, mother's education, mother's access to media, birth order and birth size |
| HH Environment | Improved source of drinking water, improved sanitation facility, water treatment, handwashing facility, no of rooms for sleeping, indoor smoking, indoor cooking, firewood cooking and floor type |

classifies them into six groups: upper caste groups, relatively advantaged janajati, religious minorities, disadvantaged non-Dalit Terai castes, disadvantaged janajati, and Dalits. For the purpose of this study, we further divide these into two broad categories: advantaged ethnic groups (comprising upper caste groups and relatively advantaged janajati) and disadvantaged groups (encompassing all others) [25].

The household environmental characteristics included water, hygiene, and sanitation practices, based on definition outlined in the DHS Statistics Guide17 and included the following: improved source of drinking water, improved sanitation facility, water treatment, handwashing facility, indoor smoking, indoor cooking, and firewood cooking [26]. Improved source of drinking water is defined as household using water from piped into dwelling, piped to yard/plot, public tap/standpipe, piped to neighbour, tube well or borehole, protected well, protected spring, rainwater, tanker truck, cart with small tank, or bottled water.

## Statistical analysis

Extraction of tree cover data for each DHS cluster followed approach described in the DHS covariate manual. It involved creation of DHS buffer of 2 km radii for urban clusters and 10 km radii for rural clusters to account for cluster displacement. Mosaic and raster clipping were conducted for the tree cover raster dataset. Water surface clouded and shadows were omitted from the raster dataset using the Generate Excluded Area function available in ArcGIS Pro. Zonal statistics was performed to extract tree cover percentage for each DHS buffer. Tree cover data was then linked to the DHS dataset for the final analysis.

We employed a logit model in the geo-linked NHDS dataset to compute the marginal effect of forest cover on ARI, diarrhoea, and fever. Logit models with clustered standard errors at the village level are well-suited to the structure of our data, which comes from repeated cross-sectional surveys where households are not repeated across DHS waves. While households are nested within villages, the clustering in our data was appropriately accounted for by adjusting standard errors at the village (i.e., the DHS cluster) level, addressing intra-cluster correlation and ensuring valid inference. All covariates were included as potential confounders in the logit models.

The baseline specification takes the following functional form:

$$P\,(\text{infection} = 1|X) = \Phi(\beta_0 + \beta_1\text{fcover} + \beta_2\text{hhc} + \beta_3\text{mch} + \beta_4\text{hhenvironment})$$

where, *infection* is a binary outcome which takes a value of "1" for infection symptoms (i.e., respiratory symptoms, diarrhoea and fever) and "0" otherwise; *fcover* is the percentage of tree cover within 2 km (Urban) and 10 km (Rural) buffer of DHS cluster; *hhc* is a vector of household socio-demographic characteristics; *mch* is a vector of maternal and child health characteristics; and *hhenvironment* is a vector of household environmental characteristics.

Predicted probabilities for each outcome variable was calculated separately for 2011 and 2016. The differences in average infection probabilities between the years 2011 and 2016 were calculated by including a dummy variable for the year 2016 in the regression model. This dummy variable takes the value of "1" for observations from the 2016 DHS wave and "0" for the 2011 wave. The coefficient of this dummy variable captures the change in the average probability of infection in 2016 relative to 2011, holding all other covariates constant. All the statistical analysis were conducted using STATA version 17.

### Ethical approval

We obtained permission to use the NDHS 2011 and 2016 datasets from The DHS Program after registering and providing the research title and objectives. Both NDHS 2011 and 2016 surveys received ethical approval from the Ethical Review Board of the Nepal Health Research Council and the Institutional Review Board of ICF International. Written informed consent was obtained from all participants prior to their enrolment in the surveys.

## Results

### Changes in prevalence of childhood illnesses and tree cover from 2011 to 2016

Between 2011 and 2016, the mean percentage of tree cover dropped from 21% to 19%. During the same period, there was an increase in the mean prevalence of fever (from 18% in 2011 to 20% in 2016), while the prevalence of diarrhoea (from 14% to 7%) and ARI symptoms (from 5% to 3%) decreased (For detailed information, please refer to Table 2).

The proportion of the population residing in rural areas decreased from 79% in 2011 to 43% in 2016. Education levels among mothers improved notably, with the average years of schooling rising from 3.63 years in 2011 to 5.01 years in 2016. Furthermore, improved water sources increased from 83% to 94% and sanitation facilities improved from 47% to 81% during the period. The proportion of households using firewood decreased from 77% to 69%, and indoor cooking and smoking also reduced substantially. Population living in the Mountain region decreased (from 19% to 9%) while population living in the Plain region increased (from 41% to 50%).

The proportion of households engaging in water treatment practices increased from 13% in 2011 to 19% in 2016. Moreover, handwashing facilities also increased from 37% to 39%. The percentage of households using firewood for cooking dropped from 77% to 69%, indoor cooking decreased from 43% to 31%, and indoor smoking reduced from 60% to 49%.

### Predicted probability of childhood infections due to forest cover

Forest cover was significantly associated with the likelihood of reduced diarrhoea symptoms in 2011 and 2016 (see Table 3). No significant association was observed between forest cover, and fever and respiratory symptoms in 2011 and 2016. From 2011 to 2016, there was a notable decrease in the likelihood of diarrhoea symptoms by 6.5% ($\Delta y/\Delta x$: −0.0646, 95% CI: −0.0504 to −0.0787), along with a 2.4% reduction ($\Delta y/\Delta x$: −0.0237, 95% CI: −0.0145 to −0.0329) in the likelihood of respiratory infections.

Change in forest cover between 2011 and 2016 was significantly associated with a reduced probability of diarrhoea by 3.39% ($\Delta y/\Delta x$: −0.0339, 95% CI: −0.0141, −0.0535; p-value: 0.001), after adjusting for all other variables. No significant associations were found between forest cover change and changes in ARI and fever prevalence among children under five.

## Discussion

In this study, we observed a significant association between forest cover and the likelihood of diarrhoea in 2011 and 2016. Changes in forest cover between 2011 and 2016 were significantly linked to a decrease in the likelihood of diarrhoea, with

**Table 2. Descriptive statistics.**

| Variables | Year 2011 (N = 5,054) | | Year 2016 (N = 4,861) | | Year 2016 – Year 2011 | |
|---|---|---|---|---|---|---|
| | Mean | Standard Error | Mean | Standard Error | Mean difference | Standard Error |
| Fever | 0.18 | 0.0054 | 0.20 | 0.0058 | 0.024 | 0.0079 |
| Diarrhoea | 0.14 | 0.0048 | 0.07 | 0.0037 | −0.065 | 0.0061 |
| Acute respiratory infection | 0.05 | 0.0030 | 0.03 | 0.0023 | −0.022 | 0.0037 |
| Tree cover percentage | 21.14 | 0.1370 | 19.34 | 0.1588 | −1.792 | 0.2089 |
| Place of residence (Rural) | 0.79 | 0.0057 | 0.43 | 0.0071 | −0.364 | 0.0091 |
| Ethnicity (Disadvantaged) | 0.60 | 0.0069 | 0.64 | 0.0069 | 0.048 | 0.0097 |
| Religion (Non-Hindus) | 0.15 | 0.0050 | 0.13 | 0.0049 | −0.013 | 0.0070 |
| Family size | 6.14 | 0.0390 | 6.20 | 0.0421 | 0.067 | 0.0571 |
| Wealth Index | 1.79 | 0.0125 | 1.84 | 0.0126 | 0.048 | 0.0177 |
| Ecological region Mountain | 0.19 | 0.005 | 0.09 | 0.004 | −0.10 | 0.007 |
| Ecological region Hill | 0.40 | 0.007 | 0.41 | 0.007 | 0.01 | 0.009 |
| Ecological region Terai | 0.41 | 0.006 | 0.50 | 0.007 | 0.088 | 0.009 |
| Child sex (Male) | 0.52 | 0.0071 | 0.53 | 0.0072 | 0.010 | 0.0100 |
| Child age (months) | 29.83 | 0.2440 | 29.59 | 0.2492 | −0.244 | 0.3475 |
| Mother's years of schooling | 3.63 | 0.0572 | 5.01 | 0.0620 | 1.376 | 0.0841 |
| Mother's access to Media | 0.58 | 0.0070 | 0.52 | 0.0072 | −0.059 | 0.0100 |
| Birth order | 2.59 | 0.0253 | 2.27 | 0.0219 | −0.324 | 0.0333 |
| Birth Size Large | 0.18 | 0.004 | 0.16 | 0.005 | −0.03 | 0.008 |
| Birth Size Average | 0.64 | 0.006 | 0.67 | 0.007 | 0.03 | 0.009 |
| Birth Size Small | 0.18 | 0.005 | 0.17 | 0.005 | −0.002 | 0.007 |
| Water source (improved) | 0.83 | 0.0053 | 0.94 | 0.0033 | 0.116 | 0.0062 |
| Sanitation facility (Improved) | 0.47 | 0.0070 | 0.81 | 0.0056 | 0.346 | 0.0090 |
| Water treatment | 0.13 | 0.0048 | 0.19 | 0.0056 | 0.055 | 0.0074 |
| Handwashing facility | 0.37 | 0.0068 | 0.39 | 0.0070 | 0.020 | 0.0098 |
| Food cooked in firewood | 0.77 | 0.0059 | 0.69 | 0.0067 | −0.082 | 0.0089 |
| Indoor cooking | 0.43 | 0.0070 | 0.31 | 0.0067 | −0.125 | 0.0096 |
| Indoor smoking | 0.60 | 0.0069 | 0.49 | 0.0072 | −0.113 | 0.0099 |

**Table 3. Predicted probability of infection symptoms due to forest cover for 2011 and 2016.**

| | Year 2011 | | | Year 2016 | | |
|---|---|---|---|---|---|---|
| | Fever | Diarrhoea | ARI | Fever | Diarrhoea | ARI |
| Forest Cover, Predicted probability | −0.0011 | −0.0618 | 0.0082 | −0.0126 | −0.0100 | 0.0049 |
| 95% CI | (−0.034, 0.032) | (−0.097, −0.026) | (−0.005, 0.021) | (−0.043, 0.018) | (−0.016, −0.002) | (−0.009, 0.019) |
| P value | 0.948 | <0.001 | 0.241 | 0.431 | 0.006 | 0.483 |
| Pseudo R2 | 0.0313 | 0.0571 | 0.0421 | 0.0146 | 0.0316 | 0.0388 |
| Correctly Classified % | 82.00 | 86.48 | 95.27 | 79.51 | 93.04 | 97.46 |
| Number of observations | 5,022 | 5,023 | 5,049 | 4,822 | 4,814 | 4,848 |

ARI, Acute respiratory infections; CI, confidence interval.

Forest cover adjusted for sociodemographic, maternal and child health characteristics and environmental covariates such as drinking water and household sanitations.

the probability of occurrence reduced after accounting for other variables. However, no significant associations were found between changes in forest cover and the prevalence of ARI and fever among children under five.

The reduced likelihood of diarrhoea prevalence due to variations in forest cover is consistent with previous studies from Cambodia and Malawi, suggesting the credibility of our study methods and findings [9,27]. Reduced likelihood of diarrhoea in the last two decades suggests improvements in child health outcomes, which may be partly attributable to Nepal's efforts to maintain and expand forest cover through community-based forest management. Additionally, the effects of public health interventions (e.g., Diarrhoeal Disease Control Program, Open Defecation Free Campaign, and sanitation and water quality improvements) through 2011 and 2016 were acknowledged but could not be controlled in the final analyses [28]. While these programs aim to prevent specific childhood infections, such childhood diarrhoeal diseases, they may be insufficient to adequately address the broader health and well-being of children and their families, highlighting the need for a more integrated, ecosystem-based approach [3,29].

The present findings suggest that forests play a preventive role in reducing the prevalence of childhood diarrhoeal diseases. Forests might play an indirect role in reducing diarrhoea prevalence by maintaining higher water quality, as forested areas often serve as natural filters, reducing runoff and preventing pollutants from contaminating water sources [11]. Forests also contribute to soil stabilization, preventing erosion and reducing the likelihood of water sources becoming contaminated by pathogens from agricultural runoff or human waste [11]. As forest cover reduces, soil instability can lead to contaminated water sources and increased exposure to pathogens [10]. This underscores the importance of adopting an ecosystem approach that incorporates multiple scales, perspectives, and acknowledges uncertainty in addressing specific child health outcomes [3,29].

As such, the preservation and expansion of forested areas should be integrated into a comprehensive public health strategy to enhance child health outcomes [9].

Our findings highlight the importance of meeting the SDG 17 (Partnerships for the Goals), which emphases the need for partnerships across sectors between governments, organizations, and communities for effective forest preservation and environmental conservation efforts and improved child health outcomes [28]. Nepal's emphasis in forest conservation and management has had important implications for public health strategies and particularly for the improvements in child health outcomes and overall community well-being. While some areas are greener, others are not, particularly some of the areas of Nepal are losing forest cover, requiring serious attention from the government [15].

The lack of significant associations between forest cover and acute respiratory infections or fever, even after adjusting for other variables, is also noteworthy. This could imply that the environmental and ecological factors influencing respiratory infections and fever differ from those affecting diarrheal disorders [12]. While forest cover may not directly impact the prevalence of these conditions, or changes in diarrhoeal disease prevalence, it can still play an indirect role through environmental factors such as air quality, water quality, housing conditions, and exposure to pollutants or smoke, which are known to influence respiratory health [30]. Moreover, other factors such as, improvements in water, sanitation and hygiene, or overall healthier household practices, over the five-year period, may also have contributed to the observed reduction in ARI prevalence between 2011 and 2016 [23,24].

## Strengths and limitations of the study

This cross-sectional analysis effectively identifies broad trends and patterns in child health and environmental factors over time, providing valuable insights into population-level changes and potential areas for public health intervention. It offers a cost-effective way to examine associations between forest cover and child health outcomes, guiding future research and policy development.

There are a few limitations to note. First, the analysis relied on cross-sectional data, which may not establish causality between outcomes and explanatory variables. Second, infections are self-reported and are subjected to bias due to over- or under-reporting of symptoms [31]. Third, DHS point displacement may have an influence on raster analysis based on assigning spatial variables to the survey participants due to positioning errors [32,33]. Fourth, the relationship between

forest cover and childhood infections may be more complex and influenced by other factors not accounted for in this study. This would suggest a way forward to use forest data from other source which includes yearly forest loss and forest gain data, combined with provincial/district level panel data on morbidity rates.

Last, this study does not include the recent data from NDHS conducted in 2022, which may have restricted our ability to capture more recent trends. Although it is noted that forest cover has not changed significantly since 2016, the lack of forest cover data for the survey year 2022 further limits our understanding of how the current forest landscape might influence childhood health outcomes. This temporal gap in data could mean that any changes in the relationship between forest cover and childhood illnesses in more recent years remain unexplored. This gap highlights the need for ongoing research that incorporates up-to-date data to better understand the evolving relationship between environmental factors like forest cover changes and public health outcomes. Additionally, future research should consider stratified analyses based on different hygiene conditions, residential locations (urban/rural), and household income levels to identify effect modifiers and populations that could benefit more from forest cover.

## Conclusion

Forest cover plays an important role to lower the risk of diarrhoea among children under 5 years. Initiatives for forest preservation should persist, with a particular focus on regions impacted by deforestation, which could reduce the prevalence of childhood diarrhoea in those areas. More research is needed to fully understand the mechanisms underlying the child health linkages of environmental programs and to guide effective policy interventions for sustainable forest management and public health improvement. Our findings should be interpreted cautiously as they may not accurately reflect individual-level dynamics regarding the effect of forest cover on child health outcomes.

## Author contributions

**Conceptualization:** Binaya Chalise, Subash Thapa.

**Data curation:** Binaya Chalise.

**Formal analysis:** Binaya Chalise.

**Investigation:** Amit Timilsina, Kedir Y Ahmed, Subash Thapa.

**Methodology:** Amit Timilsina, Binaya Chalise, Kedir Y Ahmed, Subash Thapa.

**Supervision:** Subash Thapa.

**Validation:** Kedir Y Ahmed, Subash Thapa.

**Writing – original draft:** Amit Timilsina, Subash Thapa.

**Writing – review & editing:** Amit Timilsina, Binaya Chalise, Kedir Y Ahmed, Subash Thapa.

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
