## [Decision Letter · Decision Letter 0]

6 Dec 2024

PONE-D-24-38792Effect of Forest Cover on Prevalence of Acute Respiratory Tract Infections, Diarrhoea, and Fever among Children under 5 years in NepalPLOS ONE

Dear Dr. Timilsina,

Thank you for submitting your manuscript to PLOS ONE. After careful consideration, we feel that it has merit but does not fully meet PLOS ONE’s publication criteria as it currently stands. Therefore, we invite you to submit a revised version of the manuscript that addresses the points raised during the review process.

**Academic Editor Notes:**Please address the reviewers' comments below, especially the line about editing the manuscript for more proper English. The grammatical and spelling errors were very distracting on my own reading and make the manuscript look unprofessional to the reader, though I'm sure this does not reflect the authors' efforts. I also want to highlight Reviewer 2's comments about Tables 2 and 3 - I agree that these are difficult to understand and require explanations; new organization of the tables; or both. ==============================

We look forward to receiving your revised manuscript.

Kind regards,

Andrew G Wu

Academic Editor

PLOS ONE

Journal Requirements:

Reviewers' comments:

Reviewer's Responses to Questions

**Comments to the Author**

1. Is the manuscript technically sound, and do the data support the conclusions?

Reviewer #1: Yes

Reviewer #2: Partly

2. Has the statistical analysis been performed appropriately and rigorously? 

Reviewer #1: Yes

Reviewer #2: I Don't Know

3. Have the authors made all data underlying the findings in their manuscript fully available?

Reviewer #1: Yes

Reviewer #2: No

4. Is the manuscript presented in an intelligible fashion and written in standard English?

Reviewer #1: Yes

Reviewer #2: Yes

5. Review Comments to the Author

Reviewer #1: This is an interesting study, providing a assessment of association between forest cover and infectious diseases in Nepal. Although the scale of the study is small, they provided a reasonable analysis and discussed their results well. I only have minor revision.

1. The line number should be provided in the article thoroughly.

2. The results in Table 3 should be provided as “estimate (95% CI)”.

3. What’s the criteria of covariates selection? It should be provided in method section.

4. Reference should be provided in the last paragraph in page 13.

5. Reference should be provided in the last paragraph in page 14.

6. For my consideration, it’s a cross-sectional study rather than ecological analysis.

7. I recommend authors to do stratify analysis according to different hygiene conditions, residential locations (urban/rural), and household income levels to identify effect modifiers and seek populations that could be more beneficial from forest cover.

Reviewer #2: Reviewer report PONE-D-24-38792

This is a very interesting ecological study analysing the impact of forest canopy area on the prevalence of acute respiratory infections, fever and diarrhoea using country wide population health data (census) in Nepal.

The sampling framework enabled random samples of households to be recruited that are representative of the diverse topography and geography in this country.

I have a few questions for the authors to improve clarity and my understanding of their methods and findings.

Introduction

I would like to know more about the physiological or biological link between levels of forestation and its impact on childhood fever, ARI and diarrhoea so that the research question is more easily understood.

Methods

Were all the covariates named in Table 1 used in the logit models as potential confounders?

Can the authors please explain how ethnicity was used as a measure of “advantage/disadvantage” (Page 8) – this might be obvious to the authors but I do not relate to ethnicity being analysed in this way.

Why did the authors select a “logit” (logistic regression) model rather than a “generalized linear mixed model”

Table 2: Only mean and standard error for each variable are reported – it is hard to understand the distribution of the variables. I would like to see more detailed summary statistics.

Table 3:

The lay out of Table 3 is difficult to follow – what are the numbers in the brackets for the “Forest Cover” row? Are these p-values?

The ** p<0.01 – is this correctly reported in Table 3, if the numbers in the brackets are p-values? Please clarify this.

Why not report the 95%CI in the Table 3 – it would be helpful see the range?

Why are there rows for covariates and what does “yes” mean?

The probability of lower levels of reported fever in 2016 is remarkably larger than in 2011 – am I reading the results correctly?

How was the change in probability from 2011 to 2016 calculated? I can see the formula but cannot see how the calculation was conducted.

Minor issues

There are numerous grammatical and spelling errors throughout the manuscript, although they do not detract from the meaning of the manuscript, they will need to be addressed to improve readability.

6. PLOS authors have the option to publish the peer review history of their article (what does this mean? ). If published, this will include your full peer review and any attached files.

**Do you want your identity to be public for this peer review?** For information about this choice, including consent withdrawal, please see our Privacy Policy .

Reviewer #1: No

Reviewer #2: No

---

## [Author Response · Author response to Decision Letter 1]

23 Dec 2024

22/12/2024

Dr Andrew G Wu

Academic Editor

PLOS ONE

RE: Submission of the revised manuscript, PONE-D-24-38792

Thank you for your time and feedback on our manuscript titled, Effect of Forest Cover Change on the Prevalence of Acute Respiratory Tract Infections, Diarrhoea, and Fever Among Children Under Five: Using an Ecosystem Approach to Child Health.

We have carefully revised our manuscript and hope that the manuscript has improved substantially. Should you require any additional information, please do not hesitate to contact me.

Sincerely,

Mr. Amit Timilsina, MPH

E-mail: timilsinaamit@gmail.com

Reviewers' comments

Reviewer #1

This is an interesting study, providing a assessment of association between forest cover and infectious diseases in Nepal. Although the scale of the study is small, they provided a reasonable analysis and discussed their results well. I only have minor revision.

1. The line number should be provided in the article thoroughly.

RESPONSE: Amendment made.

2. The results in Table 3 should be provided as “estimate (95% CI)”.

RESPONSE: The format of Table 3 is revised to include results as “estimate (95% CI)”.

3. What’s the criteria of covariates selection? It should be provided in method section.

RESPONSE: Covariates included in the analysis were selected based on theoretical relevance, evidence from prior literature, and data availability. This information has been included in the methods section.

4. Reference should be provided in the last paragraph in page 13.

RESPONSE: References have been added.

5. Reference should be provided in the last paragraph in page 14.

RESPONSE: References have been added.

6. For my consideration, it’s a cross-sectional study rather than ecological analysis.

RESPONSE: Thank you, the amendment has been made.

7. I recommend authors to do stratify analysis according to different hygiene conditions, residential locations (urban/rural), and household income levels to identify effect modifiers and seek populations that could be more beneficial from forest cover.

RESPONSE: We appreciate the reviewer’s suggestion to conduct stratified analyses. However, we opted not to pursue these analyses as stratification would reduce statistical power due to smaller sample sizes within subgroups. We acknowledge the value of this approach and suggest it as a direction for future research.

Reviewer #2: Reviewer report PONE-D-24-38792

This is a very interesting ecological study analysing the impact of forest canopy area on the prevalence of acute respiratory infections, fever and diarrhoea using country wide population health data (census) in Nepal.

The sampling framework enabled random samples of households to be recruited that are representative of the diverse topography and geography in this country.

I have a few questions for the authors to improve clarity and my understanding of their methods and findings.

Introduction

1. I would like to know more about the physiological or biological link between levels of forestation and its impact on childhood fever, ARI and diarrhoea so that the research question is more easily understood.

RESPONSE: Thank you for your feedback. Explanations about the physiological and ecological mechanisms linking forest cover to childhood infections have been added to the Introduction section with references.

Methods

2. Were all the covariates named in Table 1 used in the logit models as potential confounders?

RESPONSE: All covariates listed in Table 1 were included as potential confounders in the logit models. This has been mentioned in the statistical analysis section.

3. Can the authors please explain how ethnicity was used as a measure of “advantage/disadvantage” (Page 8) – this might be obvious to the authors but I do not relate to ethnicity being analysed in this way.

RESPONSE: The categorization of caste/ethnicity in DHS data for Nepal is based on the caste system, which includes more than 125 groups. Due to the socioeconomic status and cultural hierarchy of these castes and ethnicities in Nepalese society, the DHS classifies them into six groups: upper caste groups, relatively advantaged janajati, religious minorities, disadvantaged non-Dalit Terai castes, disadvantaged janajati, and Dalits. For the purpose of this study, we further divide these into two broad categories: advantaged ethnic groups (comprising upper caste groups and relatively advantaged janajati) and disadvantaged groups (encompassing all others).

4. Why did the authors select a “logit” (logistic regression) model rather than a “generalized linear mixed model”

RESPONSE: We used logistic regression with clustered standard errors at the village level because it is well-suited to the structure of our data, which comes from repeated cross-sectional surveys where households are not repeated across DHS waves. While households are nested within villages, the clustering in our data was appropriately accounted for by adjusting standard errors at the village (i.e. the DHS cluster) level, addressing intra-cluster correlation and ensuring valid inference. GLMMs are typically advantageous when analyzing hierarchical data with repeated measures or when random effects can capture substantial within-cluster variability. However, in this case, the absence of repeated measures for households and the cross-sectional nature of the data makes random effects unnecessary. Logistic regression with clustered standard errors provides a parsimonious and interpretable approach, aligning with our objective to assess population-level relationships while appropriately accounting for the data structure.

5. Table 2: Only mean and standard error for each variable are reported – it is hard to understand the distribution of the variables. I would like to see more detailed summary statistics.

RESPONSE: Thank you. We acknowledge the value of additional statistics however, we opted to present the means and standard errors as they align with the primary objectives of our analysis, which focus on comparative trends over time rather than detailed distributional characteristics, which are in fact available in Demographic Survey reports.

Table 3:

6. The lay out of Table 3 is difficult to follow – what are the numbers in the brackets for the “Forest Cover” row? Are these p-values?

RESPONSE: Table 3 has been revised for clarity.

7. The ** p<0.01 – is this correctly reported in Table 3, if the numbers in the brackets are p-values? Please clarify this.

RESPONSE: Amendment made.

8. Why not report the 95%CI in the Table 3 – it would be helpful see the range?

RESPONSE: The 95% confidence intervals of the predicted probabilities are added to Table 3 for all estimates.

9. Why are there rows for covariates and what does “yes” mean?

RESPONSE: Clarified in Table 3.

10. The probability of lower levels of reported fever in 2016 is remarkably larger than in 2011 – am I reading the results correctly?

RESPONSE: Yes you are correct.

11. How was the change in probability from 2011 to 2016 calculated? I can see the formula but cannot see how the calculation was conducted.

RESPONSE: The differences in average infection probabilities between the years 2011 and 2016 were calculated by including a dummy variable for the year 2016 in the regression model. This dummy variable takes the value of "1" for observations from the 2016 DHS wave and "0" for the 2011 wave. The coefficient of this dummy variable captures the change in the average probability of infection in 2016 relative to 2011, holding all other covariates constant. Specifically, the regression equation in page 7 can be expressed as:

P(infection=1│X)= Φ(β_0+β_1 fcover+β_2 DHS2016+⋯)

Here, β_2 DHS2016 quantifies the difference in the baseline log-odds of infection between 2016 and 2011, controlling for forest cover, household socio-demographic characteristics, maternal and child health characteristics, and household environmental characteristics. The corresponding probability difference can be computed by transforming the predicted log-odds into probabilities using the inverse of the link function (in this case, the cumulative normal distribution, Φ).

Minor issues

12. There are numerous grammatical and spelling errors throughout the manuscript, although they do not detract from the meaning of the manuscript, they will need to be addressed to improve readability.

RESPONSE: The manuscript has undergone a thorough proofreading process to address grammatical and spelling errors.

---

## [Decision Letter · Decision Letter 1]

7 Mar 2025

PONE-D-24-38792R1Effect of Forest Cover Change on the Prevalence of Acute Respiratory Tract Infections, Diarrhoea, and Fever Among Children Under Five: Using an Ecosystem Approach to Child HealthPLOS ONE

Dear Dr. Timilsina,

Thank you for submitting your manuscript to PLOS ONE. After careful consideration, we feel that it has merit but does not fully meet PLOS ONE’s publication criteria as it currently stands. Therefore, we invite you to submit a revised version of the manuscript that addresses the points raised during the review process. Please go through the Reviewers comments in detail and respond to them accordingly.

We look forward to receiving your revised manuscript.

Kind regards,

Furqan Kabir

Academic Editor

PLOS ONE

Reviewers' comments:

Reviewer's Responses to Questions

**Comments to the Author**

1. If the authors have adequately addressed your comments raised in a previous round of review and you feel that this manuscript is now acceptable for publication, you may indicate that here to bypass the “Comments to the Author” section, enter your conflict of interest statement in the “Confidential to Editor” section, and submit your "Accept" recommendation.

Reviewer #2: (No Response)

Reviewer #3: (No Response)

2. Is the manuscript technically sound, and do the data support the conclusions?

Reviewer #2: Yes

Reviewer #3: Yes

3. Has the statistical analysis been performed appropriately and rigorously? 

Reviewer #2: Yes

Reviewer #3: Yes

4. Have the authors made all data underlying the findings in their manuscript fully available?

Reviewer #2: Yes

Reviewer #3: Yes

5. Is the manuscript presented in an intelligible fashion and written in standard English?

Reviewer #2: Yes

Reviewer #3: Yes

6. Review Comments to the Author

Reviewer #2: I thank the authors for responding to the reviewers’ comments and questions. Aside from some minor grammar edits, I recommend this manuscript be accepted for publication. I look forward to future research which tracks the trends identified in these analyses.

Grammar/edits

Line 95: ‘sub-region’ should be ‘sub-regions’

Line 96: ‘province’ should be ‘provinces’

Line 141: please add a comma after ‘dwelling’

Line 111: ArcGIS Pro requires some reference with the version number and manufacturer.

Line 180: Please consider adding “decreased” after “(5% to 3%) as the sentence is incomplete.

Line 180: Please edit the last sentence to “For detailed information, please refer to Table 2”.

Reviewer #3: Thank you for the revised manuscript. Most of the comments from the first review were addressed. however- I still feel the following points still need to be addressed.

Page 3 line 50 – Are these global statistics? Please specify

PAGE 8 LINES 178- 180 The second part of the sentence seems incomplete

Page 11 lines 244-245 While I understand this reasoning, I do not think that forest cover is directly related to diarrhoea either.

One of the challenges in understanding the results is the lack of sample statistics (n +%). This helps us understand the prevalence of the different conditions and could also help us understand the non-significant results. Although the authors provide reasons for presenting means and standard errors only in their response- including sample distribution could still enhance the interpretation of the results.

7. PLOS authors have the option to publish the peer review history of their article (what does this mean? ). If published, this will include your full peer review and any attached files.

**Do you want your identity to be public for this peer review?** For information about this choice, including consent withdrawal, please see our Privacy Policy .

Reviewer #2: No

Reviewer #3: No

---

## [Author Response · Author response to Decision Letter 2]

19 Mar 2025

13/03/2025

Dr Furqan Kabir

Academic Editor

PLOS ONE

RE: Submission of the revision 2 of the manuscript, PONE-D-24-38792

Thank you for your time and feedback on our manuscript titled, Effect of Forest Cover Change on the Prevalence of Acute Respiratory Tract Infections, Diarrhoea, and Fever Among Children Under Five: Using an Ecosystem Approach to Child Health.

We have carefully revised our manuscript based on the feedback. Should you require any additional information, please do not hesitate to contact me.

Sincerely,

Mr. Amit Timilsina, MPH

E-mail: timilsinaamit@gmail.com

Reviewers' comments

Reviewer #2

I thank the authors for responding to the reviewers’ comments and questions. Aside from some minor grammar edits, I recommend this manuscript be accepted for publication. I look forward to future research which tracks the trends identified in these analyses.

Grammar/edits

Line 95: ‘sub-region’ should be ‘sub-regions’

Line 96: ‘province’ should be ‘provinces’

Line 141: please add a comma after ‘dwelling’

Line 111: ArcGIS Pro requires some reference with the version number and manufacturer.

Line 180: Please consider adding “decreased” after “(5% to 3%) as the sentence is incomplete.

Line 180: Please edit the last sentence to “For detailed information, please refer to Table 2”.

RESPONSE: Thank you. All the suggested amendments have been made.

Reviewer #3

Thank you for the revised manuscript. Most of the comments from the first review were addressed. however- I still feel the following points still need to be addressed.

Page 3 line 50 – Are these global statistics? Please specify

RESPONSE: Amendment made.

PAGE 8 LINES 178- 180 The second part of the sentence seems incomplete

RESPONSE: Amendment made.

Page 11 lines 244-245 While I understand this reasoning, I do not think that forest cover is directly related to diarrhoea either.

RESPONSE: Thank you. The paragraph has been revised for clarity.

One of the challenges in understanding the results is the lack of sample statistics (n +%). This helps us understand the prevalence of the different conditions and could also help us understand the non-significant results. Although the authors provide reasons for presenting means and standard errors only in their response- including sample distribution could still enhance the interpretation of the results.

RESPONSE: Thank you for your comment. We truly appreciate your time and feedback. However, we prefer to maintain a focused presentation in this paper on the primary trends and the implications of forest cover changes on childhood infections. As you know, there have been published numerous papers over the past two decades on childhood fever, ARI, and diarrhoea using DHS data, focusing on the detailed demographic and distributional context. We hope you will agree with our rationale and decision. Thank you once again.

---

## [Editor Report · Decision Letter 2]

21 Mar 2025

PONE-D-24-38792R2Effect of Forest Cover Change on the Prevalence of Acute Respiratory Tract Infections, Diarrhoea, and Fever Among Children Under Five: Using an Ecosystem Approach to Child HealthPLOS ONE

Dear Dr. Timilsina,

Thank you for submitting your manuscript to PLOS ONE. After careful consideration, we feel that it has merit but does not fully meet PLOS ONE’s publication criteria as it currently stands. Therefore, we invite you to submit a revised version of the manuscript that addresses the points raised during the review process.

There are few minor comments that really need your attention, please review and respond accordingly.

We look forward to receiving your revised manuscript.

Kind regards,

Furqan Kabir

Academic Editor

PLOS ONE
---

## [Author Response · Author response to Decision Letter 3]

11 Apr 2025

11/04/2025

Dr Furqan Kabir

Academic Editor

PLOS ONE

RE: Submission of the revision 3 of the manuscript, PONE-D-24-38792

Thank you for your feedback on our manuscript titled, Effect of Forest Cover Change on the Prevalence of Acute Respiratory Tract Infections, Diarrhoea, and Fever Among Children Under Five: Using an Ecosystem Approach to Child Health. We have revised the reference list based on your comment.

Sincerely,

Mr. Amit Timilsina, MPH

E-mail: timilsinaamit@gmail.com

Journal Requirements:

1. Please include your full ethics statement in the ‘Methods’ section of your manuscript file. In your statement, please include the full name of the IRB or ethics committee who approved or waived your study, as well as whether or not you obtained informed written or verbal consent. If consent was waived for your study, please include this information in your statement as well.

RESPONSE: Thank you. Ethics statement included.

RESPONSE: Thank you. We have carefully checked all the references. References 1, 2, 4, 17, 23, and 24 have been corrected.

---

## [Editor Report · Decision Letter 3]

15 Apr 2025

PONE-D-24-38792R3Effect of Forest Cover Change on the Prevalence of Acute Respiratory Tract Infections, Diarrhoea, and Fever Among Children Under Five: Using an Ecosystem Approach to Child HealthPLOS ONE

Dear Dr. Timilsina,

Thank you for submitting your manuscript to PLOS ONE. After careful consideration, we feel that it has merit but does not fully meet PLOS ONE’s publication criteria as it currently stands. Therefore, we invite you to submit a revised version of the manuscript that addresses the points raised during the review process.

Please go through the Reviewers comments and draft your responses accordingly.

We look forward to receiving your revised manuscript.

Kind regards,

Furqan Kabir

Academic Editor

PLOS ONE

---

## [Author Response · Author response to Decision Letter 4]

10 May 2025

24/04/2025

Maidelyn R. Peregrin

PLOS ONE

RE: Submission of the revision 6 of the manuscript, PONE-D-24-38792

Thank you for your feedback regarding the use of figures containing map-based data. After careful consideration, we have decided to remove Figures 1 and 2 from the manuscript.

Sincerely,

Mr. Amit Timilsina, MPH

E-mail: timilsinaamit@gmail.com

Journal Requirements:

1. 1. We note that Figures 1,2 and 3 in your submission contain [map/satellite] images which may be copyrighted. All PLOS content is published under the Creative Commons Attribution License (CC BY 4.0), which means that the manuscript, images, and Supporting Information files will be freely available online, and any third party is permitted to access, download, copy, distribute, and use these materials in any way, even commercially, with proper attribution. For these reasons, we cannot publish previously copyrighted maps or satellite images created using proprietary data, such as Google software (Google Maps, Street View, and Earth). For more information, see our copyright guidelines: http://journals.plos.org/plosone/s/licenses-and-copyright.

1. You may seek permission from the original copyright holder of Figures 1,2 and 3 to publish the content specifically under the CC BY 4.0 license.

Natural Earth (public domain): http://www.naturalearthdata.com/"

RESPONSE: Thank you for your feedback regarding the use of figures containing map-based data. After careful consideration, we have decided to remove Figures 1 and 2 from the manuscript.

Our study primarily aims to examine the relationship between forest cover change and the risk of common childhood infections among children under five in Nepal. While spatial representation of forest cover was initially included for context, it is not central to our research objectives or analyses. We believe that removing these figures will not affect the clarity or scientific contribution of the study.

We appreciate your guidance and hope this revised version aligns better with the journal’s licensing requirements.

---

## [Editor Report · Decision Letter 4]

28 Jul 2025

Effect of Forest Cover Change on the Prevalence of Acute Respiratory Tract Infections, Diarrhoea, and Fever Among Children Under Five: Using an Ecosystem Approach to Child Health

PONE-D-24-38792R4

Dear Dr. Timilsina,

We’re pleased to inform you that your manuscript has been judged scientifically suitable for publication and will be formally accepted for publication once it meets all outstanding technical requirements.

Kind regards,

Keiko Nakamura

Academic Editor

PLOS ONE